# Prognostic Role of Neutrophil Percentage-to-Albumin Ratio in Patients with Non-ST-Elevation Myocardial Infarction

**DOI:** 10.3390/medicina60122101

**Published:** 2024-12-22

**Authors:** Mehmet Karaca, Ayca Gumusdag

**Affiliations:** 1Cardiology Department, Atasehir Memorial Hospital, Uskudar University, Istanbul 34758, Turkey; 2Dr. Siyami Ersek Thoracic and Cardiovascular Surgery Education Research Hospital, University of Health Sciences Turkey, Istanbul 34668, Turkey; aycagumusdag999@hotmail.com

**Keywords:** inflammation, neutrophil percentage, serum albumin, C-reactive protein, NPAR, non-ST elevation myocardial infarction, adverse outcome

## Abstract

*Background and Objectives*: This study aimed to investigate whether neutrophil percentage-to-albumin ratio (NPAR) levels on admission have prognostic significance regarding one-year major adverse cardiovascular and cerebrovascular events (MACCEs) in non-ST-elevation myocardial infarction (NSTEMI) patients. *Materials and Methods*: A total of 464 patients aged 59.2 ± 11.6 years constituted the cohort of this retrospectively designed study. Considering a 1-year follow-up period, the patients were divided into two groups: those with MACCEs and those without. The complete blood count, serum C-reactive protein and serum albumin levels were measured at admission. The NPAR, C-reactive protein/albumin ratio (CAR) and systemic immune-inflammation (SII) index were calculated for all patients, and the associations of these inflammatory-based biomarkers with 1-year MACCEs were evaluated. *Results*: During the 12-month follow-up period, MACCEs were observed in 75 (16.2%) patients, of which 35 (7.5%) patients died. The patients with MACCEs had higher CRP (*p* < 0.001), a higher percentage of neutrophils (*p* < 0.001), lower albumin levels (*p* < 0.001), a higher CAR (*p* < 0.001), a higher SII index (*p* = 0.008) and a higher NPAR (*p* < 0.001). A high anatomical SxSI score, a high low-density lipoprotein cholesterol level, hypoalbuminemia, high neutrophil counts, a high NPAR level and a high CAR level were independent predictors for one-year MACCEs (all *p* < 0.05). The NPAR (area under the curve [AUC] = 0.775, *p* < 0.001) and albumin level (AUC = 0.708, *p* < 0.001) had better and sufficient discriminatory power and predictive accuracy in determining one-year MACCEs, when compared to the neutrophil (AUC = 0.693, *p* < 0.001), CAR (AUC = 0.639, *p* < 0.001) and SII index (AUC = 0.660, *p* < 0.001), in terms of the receiver operating characteristic curve. The DeLong test revealed that the predictive performance of the NPAR was superior to that of the other inflammatory parameters. In particular, individuals with an NPAR value greater than 17.6 were at greater risk of developing MACCEs (*p* < 0.001). *Conclusions*: The NPAR can be used as a newly identified promising inflammatory biomarker to predict one-year MACCEs in NSTEMI patients undergoing revascularization therapy.

## 1. Introduction

Significant advances in the recognition and treatment of non-ST-elevation myocardial infarction (NSTEMI) have led to improvements in more favorable short-term clinical outcomes with 30-day mortality, with the same rate as seen in ST-elevation myocardial infarction (STEMI) patients. However, despite these significant advances in interventional and medical treatment approaches, some studies have suggested that long-term outcomes in patients with NSTEMI do not improve at the same rate as seen in STEMI patients and that 1-year mortality rates for NSTEMI are significantly higher when compared to STEMI [1,2]. Therefore, adequate risk stratification and comprehensive clinical follow-up of NSTEMI patients is expected to allow for more precise therapeutic management of these patients, thus avoiding adverse outcomes.

As the prognosis of NSTEMI patients is typically poor, it is vital to identify prognostically important markers. Recent evidence has revealed that systemic inflammation is associated with the development of atherosclerosis, changes in hemostasis, presence and severity of coronary artery disease (CAD), acute coronary thrombosis and future adverse coronary events [3,4]. The main mechanisms associated with clinical instability in NSTEMI patients are the degree of inflammatory activity, increased vasomotor tone and plaque activation [4,5]. In addition, recent evidence has shown that various inflammatory biomarkers, such as high-sensitivity C-reactive protein (hs-CRP), the systemic immune-inflammation (SII) index, the advanced lung cancer inflammation index (ALI) and the C-reactive protein-to-albumin ratio (CAR) are closely related to the prognosis of patients suffering from acute coronary syndrome (ACS) [4,6,7,8,9]. Thus, taking inflammatory markers into consideration in risk assessment may guide better prediction for prognosis and treatment planning in NSTEMI patients. In fact, recent evidence has shown that inflammation markers are useful in evaluating the prognosis of heart disease, and data indicate that it is possible to take these markers into account in risk assessment, potentially even increasing the prognostic value of risk scores by including them in relevant scoring systems [4].

The neutrophil percentage-to-albumin ratio (NPAR) is a newly defined inflammatory biomarker that has been found to be associated with adverse clinical outcomes in several cardiovascular diseases (CVDs), such as chronic coronary artery disease, STEMI, heart failure, stroke and cardiogenic shock [10,11,12,13,14,15,16,17]. Therefore, based on the literature, we investigated the impact of NPAR levels at hospital admission on one-year cardiovascular outcomes in NSTEMI patients undergoing percutaneous coronary intervention (PCI), in comparison with other inflammatory-based markers (including the CAR, ALI and SII index) whose predictive value has previously been demonstrated in this regard. To the best of our knowledge, this is the first study on this topic.

## 2. Material and Methods

The medical records of consecutive patients admitted to the Department of Cardiology, Dr. Siyami Ersek Thoracic and Cardiovascular Surgery Education Research Hospital, University of Health Sciences Turkey, and Atasehir Memorial Hospital, Uskudar University, were retrospectively reviewed between January 2017 and December 2023. Data collected within the scope of routine clinical practice and compiled in the database were examined. Figure 1 presents a detailed flowchart showing the participant selection process and the study exclusion criteria.

The study cohort consisted of 464 patients who were definitively diagnosed with NSTEMI and underwent coronary angiography with PCI for the atherosclerotic de novo culprit lesion. The patients’ clinical and demographic history, including body mass index (BMI), hypertension (HT), diabetes (DM), family history, hyperlipidemia, smoking and vascular diseases—defined as previous myocardial infarction (MI), peripheral arterial disease (PAD), ischemic stroke or transient ischemic attack history—were obtained by reviewing the electronic medical records of patients. PAD was defined as atherosclerotic disease in arteries other than the coronaries and aorta, with exercise-induced claudication, a history of revascularization therapy, the presence of decreased or absent pulse on physical examination, a history of amputation or stenosis greater than 50% on angiography [18]. DM was defined as a fasting glycemia > 126 mg/dL and random glycemia > 200 mg/dL [19]. HT was defined as a resting systolic blood pressure of more than 140 mmHg and/or diastolic blood pressure > 90 mmHg on at least two occasions in the office [20]. Hyperlipidemia was defined as an elevated fasting total cholesterol concentration (≥200 mg/dL) and/or a low-density lipoprotein cholesterol level (≥130 mg/dL), according to the National Cholesterol Education Program-3 recommendations [21]. Active cigarette-smoking status was defined as smoking more than 10 cigarettes per day for at least one year without an attempt to quit. Family history was defined as the presence of heart disease or sudden cardiac death in a male first-degree relative aged younger than 55 years old or in a female first-degree relative aged younger than 65 years old. Chronic heart failure (HF) was defined as a history of HF signs and symptoms confirmed using objective evidence supporting cardiac dysfunction or a left ventricular ejection fraction (LVEF) of less than 40%. Chronic renal failure (CRF) was defined as the presence of kidney damage (albuminuria of at least 30 mg per 24 h, hematuria or structural abnormalities such as polycystic or dysplastic kidneys) or an estimated glomerular filtration rate (eGFR) below 60 mL/min/1.73 m^2^, regardless of the cause, lasting 3 months or longer [22]. In addition, physical examination findings for the entire study population were obtained.

A diagnosis of NSTEMI was made according to symptoms, electrocardiographic findings and other accessory examinations, with the standard of NSTEMI diagnosis based on the diagnostic criteria of NSTEMI in the European Society of Cardiology and American College of Cardiology/American Heart Association guidelines [23,24]. Presentation with acute chest pain or overwhelming shortness of breath accompanied by elevated cardiac troponin without persistent ST segment elevation is suggestive of NSTEMI, except in patients with true posterior myocardial infarction. All patients were treated in accordance with the current guidelines.

Routine complete blood cell counts, including the levels of white blood cells, neutrophils, lymphocytes, hemoglobin and platelets, were evaluated using an automated blood cell counter. In addition, blood evaluations for the determination of the blood glucose, creatinine, uric acid, C-reactive protein, serum albumin, total cholesterol, low-density lipoprotein cholesterol (LDL-C), high-density lipoprotein cholesterol (HDL-C), triglycerides and electrolytes were analyzed using venous blood samples obtained at admission. The blood parameters were determined by the clinical laboratories of the Dr. Siyami Ersek Thoracic and Cardiovascular Surgery Education Research Hospital and Atasehir Memorial Hospital.

The NPAR was calculated as the ratio of neutrophil percentage to the albumin level in the blood samples collected at admission, according to the following formula: [Neutrophils percentage (%) × 100/Albumin (g/dL)]. The C-reactive protein-to-albumin ratio (mg/g) was calculated as the ratio of CRP (mg/dL) to the serum albumin (g/dL) concentration. The SII index was calculated as platelet count × neutrophil count/lymphocyte count. The ALI was calculated according to the patient’s BMI, serum albumin level and neutrophil-to-lymphocyte ratio (NLR), as ALI = BMI × serum albumin/NLR.

The angiograms of the patients were evaluated by two experienced interventional cardiologists who were blinded to our study. The anatomic severity of coronary stenosis was quantitatively evaluated with the anatomical Syntax Score I (SxSI) using the downloaded version from https://syntaxscore.org (access date: 15 January 2023). The clinical anatomical Syntax Score PCI was also calculated quantitatively using the same website.

The primary endpoint of the present study was MACCEs, defined as cardiovascular death (comprising myocardial infarction, significant cardiac arrhythmia, heart failure and any stroke), non-fatal myocardial infarction, non-fatal cerebrovascular accident (ischemic stroke or TIA) or cardiovascular re-hospitalization due to myocardial infarction, significant cardiac arrhythmia, heart failure or any stroke within the 12-month follow-up period, based on the Academic Research Consortium-2 consensus [25]. The study population was divided into two groups, according to whether or not MACCEs occurred.

As our study was retrospectively designed, written informed consent from participants could not be obtained; however, our study protocol was approved by the ethics committee of the Dr. Siyami Ersek Thoracic and Cardiovascular Surgery Education Research Hospital, University of Health Sciences (Protocol number: E-28001928-604.01-256386451).

## 3. Statistical Analysis

The normally distributed continuous variables are expressed as the means ± standard deviations, while those not normally distributed are expressed as medians with interquartile ranges. The categorical variables are reported as percentages. The chi-squared (χ^2^) test was used to compare the categorical variables between the groups. The Kolmogorov–Smirnov test was used to assess whether the variables were normally distributed. Student’s *t*-test or the Mann–Whitney U-test were used to compare the continuous variables between the groups, according to whether they were normally distributed or not. Pearson coefficients were used to define the degree of correlation of normally distributed parameters, and Spearman’s correlations were used for non-normally distributed parameters. In order to determine the independent predictors of MACCEs, variables associated with a *p* < 0.05 level, according to the univariate analysis, were included in the multivariate cox regression analysis, with the results reported as hazard ratios (HR) and 95% confidence intervals (CI). To determine the predictive accuracy and discriminative capacity of the NPAR, CAR and SII index in determining the MACCEs, the receiver operating characteristic (ROC) curve and area under the curve (AUC), accompanied by the 95% confidence interval, was obtained. Discriminatory power was classified as good if AUC ≥ 0.70 and inadequate if AUC < 0.70 [26]. To compare the predictive performance of the NPAR, neutrophil, albumin and other inflammation-based scores in determining the development of one-year MACCEs, pairwise comparisons of the ROC curves were performed using the method of DeLong et al. [27]. ROC analysis was also conducted to evaluate the sensitivity and specificity of NPAR for MACCEs, as well as to determine the optimal cut-off value using Youden’s index [28]. The time-to-event data are presented graphically with Kaplan–Meier survival curves and log-rank tests. The threshold of statistical significance was established at *p* < 0.05. All statistical analyses were performed using the Statistical Package for the Social Sciences version 24.0 software program (IBM Corp., Armonk, NY, USA). The ROC curves of the models were compared using the MEDCALC software program (MedCalc Software bv, Ostend, Belgium).

## 4. Results

A total of 464 patients diagnosed with NSTEMI with a mean age of 59.2 ± 11.6 years constituted the study cohort. During the 12-month follow-up period, among the 75 patients (16.2%) who developed MACCEs, 35 (7.5%) patients died, 18 patients suffered from non-fatal MI (3.9%), 2 (0.4%) patients had stroke and 20 (4.3%) patients were re-hospitalized for cardiovascular causes, including unstable angina (n = 10, 2.2%), decompensated heart failure (n = 7, 1.5%) and hemodynamically significant ventricular arrhythmia (n = 3, 0.6%). Additionally, 32 patients (42.7%) with MACCEs required re-vascularization therapy.

There were no significant differences between the patients with and without MACCEs in terms of their demographic characteristics and cardiovascular disease risk factors, including age, male gender, hypertension, diabetes, smoking, family history and being overweight or obese (*p* > 0.005 for all), other than hyperlipidemia (*p* = 0.004). On the other hand, patients who suffered from MACCEs had a lower LVEF (46.4 ± 7.7 vs. 48.8 ± 7.6; *p* = 0.014) and a higher GRACE risk score (111.2 ± 21.2 vs. 102.7 ± 24.4; *p* = 0.005) than those without MACCEs. Additionally, the complexity of coronary artery disease, as defined by the anatomical SxSI, was greater in patients who developed MACCEs than in others (median of 19.0 vs. median of 12.0; *p* < 0.001). Considering the biochemical and hematological parameters, the patients with MACCEs had higher peak troponin I (median of 280.0 vs. 199.0, *p* = 0.040), higher CRP (median of 6.50 vs. median of 5.10, *p* < 0.001), higher total cholesterol (221.4 ± 43.2 vs. 204.7 ± 47.9, *p* = 0.005), higher low-density lipoprotein cholesterol (147.0 ± 41.1 vs. 129.4 ± 39.1, *p* < 0.001), a higher neutrophil count (6.14 ± 1.2 vs. 5.14 ± 1.6, *p* < 0.001), a higher neutrophil percentage (68.3 ± 10.2 vs. 58.1 ± 13.5, *p* < 0.001) and lower albumin levels (3.46 ± 0.38 vs. 3.79 ± 0.45, *p* < 0.001) compared to those without MACCEs. Furthermore, in terms of inflammation-based scores, patients who suffered from MACCEs had a higher CAR (median of 1.76 vs. median of 1.33, *p* < 0.001), SII index (median of 899.2 vs. median of 633.2, *p* = 0.008), NPAR levels (19.7 ± 3.8 vs. 15.6 ± 3.9, *p* < 0.001) and a lower ALI (median of 24.5 vs. median of 37.7, *p* < 0.001) than those without MACCEs. Detailed demographics, clinical parameters and laboratory parameters of all study participants, including a comparison between the two groups, are summarized in Table 1 and Table 2. In order to determine the independent predictors of MACCEs, we performed multivariable cox regression analysis by including variables that exhibited statistically significant relationships in the univariate analysis. As albumin and neutrophils are common parameters in inflammation-based scores, four separate analysis models were constructed, either without (Model 1) or with (Model 2–4) inflammation-based scores. Furthermore, the peak troponin I level, which is a marker of cardiac injury, was weakly positively correlated with the NPAR (r = 0.161, *p* = 0.001) and the ALI (r = −0.97, *p* = 0.037), but not correlated with the CAR (r = −0.023, *p* = 624) or SII (r = 0.082, *p* = 0.08). For this reason, it was included in all regression analysis models. A high anatomical SxSI score (in all models), a high low-density lipoprotein cholesterol level (in all models), hypoalbuminemia (in Model 1), high neutrophil counts (in Model 1), increased baseline CRP levels (in Model 4), a high NPAR level (in Model 2), a high CAR level (in Model 3) and a high ALI (in Model 4) were found to be independent predictors of one-year MACCEs, according to the multivariable cox regression analysis (Table 3 and Table 4). On the other hand, the SII index did not predict the development of one-year MACCEs. Moreover, ROC curve analysis revealed that the NPAR (AUC = 0.775, *p* < 0.001) and albumin (AUC = 0.708, *p* < 0.001) had better and sufficient discriminatory power and predictive accuracy in determining one-year MACCEs compared to neutrophil (AUC = 0.693, *p* < 0.001), the CAR (AUC = 0.639, *p* < 0.001) and the SII index (AUC = 0.660, *p* < 0.001). It was also found, in the comparative analysis of the ROC curves using the DeLong test, that the discriminative ability of the NPAR was significantly superior to that of the others (Figure 2). Furthermore, NPAR values greater than 17.6 were determined as the cut-off value using the Youden index, with a 79% sensitivity and 70% specificity for the prediction of MACCEs. The Kaplan–Meier curves represented that high-risk patients with NPAR values greater than 17.6 had significantly worse prognosis and adverse events than those in the low-risk group during the follow-up period after their initial hospitalization (*p* = 0.001; Figure 3).

## 5. Discussion

The main findings of the present study are as follows: (i) A higher LDL level, the presence of severe CAD (defined by SxSI), a higher neutrophil count, hypoalbuminemia, an increased baseline CRP level, a higher NPAR value, a higher CAR value and a lower ALI value were independent predictors of one-year MACCEs in NSTEMI patients. (ii) Among these predictors, NPAR and hypoalbuminemia had sufficient predictive ability and discrimination power. However, NPAR was found to be a superior marker with statistically better predictive capability than hypoalbuminemia and other inflammatory parameters. (iii) Individuals with high NPAR levels had a worse prognosis, according to the Kaplan–Meier curve, than those without.

NSTEMI has a multi-factorial pathogenesis consisting of complex and various local and systemic changes in inflammation and immune system activity [29]. It has also been suggested that inflammation is associated with clinical instability in NSTEMI patients [4,30]. As an important part of the inflammatory process, neutrophils—beyond their contribution to the development of coronary atherosclerosis—play an important role in the development of atherothrombotic events, poor remodeling, death and composite adverse cardiac events after the index event in ACS patients [13,31]. Albumin has always been considered as an indicator of nutritional status. It is also known that a low serum albumin concentration is an indicator of the severity of inflammation and contributes to the development of CVDs [10,15]. Moreover, hypoalbuminemia has been found to be associated with mortality and poor prognosis in patients suffering myocardial infarction [10,13]. Therefore, it is not surprising that, in this study, we found that neutrophil counts were higher and albumin levels were lower in patients who developed MACCEs than in those who did not, consistent with the data in the existing literature.

As a combination of the two inflammation-related indicators mentioned above, the newly identified NPAR reflects aspects of malnutrition and inflammation. It has also been postulated that NPAR better reflects the inflammatory response and provides a more comprehensive assessment of inflammation [11,32]. In fact, there are literature data suggesting that neutrophil counts vary among individuals and may not accurately reflect the degree of inflammation. Therefore, the neutrophil percentage can provide a better reflection of inflammation by eliminating individual differences [33]. As such, the NPAR was used instead of the neutrophil-to-albumin ratio to determine the degree of inflammation in this study. In recent years, studies have shown that the NPAR is an inflammation-based prognostic predictor of various CVDs, including atrial fibrillation, chronic heart failure, coronary acute STEMI and cardiogenic shock [11,12,13,14,32,33]. In a study conducted by Cui et al., NPAR was found to be an independent predictor of in-hospital mortality in patients with STEMI [12]. Sun et al. showed that a higher NPAR level was significantly associated with higher 30-, 90- and 365-day all-cause mortality in critically ill patients with documented CAD [15]. They also suggested that, although both the neutrophil percentage and albumin level had an impact on prognosis, the NPAR had better predictive value. In another study, Wang et al. suggested that the NPAR, in addition to being associated with all-cause mortality in patients with chronic heart failure, provided a better predictive value than albumin or the neutrophil percentage alone [11]. In this study, although albumin and neutrophil predicted the development of MACCEs, consistent with previous studies, the NPAR showed a better predictive performance than either marker alone.

Some easily obtained laboratory indices of inflammation, such as the CAR, ALI and SII index, have recently been examined in the context of various cardiovascular diseases and have been shown to predict short- or long-term mortality [6,7,8,9]. Among these inflammation-based parameters, the CAR was an independent predictor of MACCEs, while the SII index was not predictive in this regard. To the best of our knowledge, there are few studies evaluating the importance of the CAR in predicting CAD severity and in-hospital mortality in NSTEMI patients [8,34]. In a study conducted by Menekse et al., the CAR was found to be a predictor for in-hospital mortality in NSTEMI patients undergoing PCI [34]. It was the first study to investigate the effect of the CAR on long-term cardiovascular outcomes. When our results are evaluated together with the literature data, we suspect that the CAR may have a prognostic role in short- and long-term events in NSTEMI patients. In addition, the current study demonstrated the prognostic role of the ALI in patients with acute coronary syndromes undergoing PCI, similar to previous studies conducted by Trimarchi et al. and Wang et al. [7,35]. There are also data regarding the prognostic importance of the SII index—another inflammatory-based parameter—in NSTEMI patients, but the obtained results are somewhat confusing [36,37]. In a study conducted by Cayırlı et al., it was shown to be a useful index in predicting all-cause mortality in the NSTEMI patient group, but the discrimination ability of the SII index was inadequate, with an AUC value of 0.650 [36]. In the current study, the SII index was not predictor of 1-year MACCEs in NSTEMI patients. When we consider the predictive performance and discriminative power of all inflammatory parameters considered in our study, it was observed that the NPAR was superior to both its components (i.e., albumin and neutrophil) and other inflammatory-based markers, including the CAR, ALI and SII index.

Although some previous studies have shown that low LDL-C levels at admission were associated with an increased risk of mortality following acute myocardial infarction (lipid paradox) [38], in line with the literature, high LDL-C levels at admission were also found to be an independent predictor of the development of 1-year adverse cardiovascular events in the present study. In addition, consistent with several previous studies, the Syntax score was found to be associated with long-term prognosis in NSTEMI patients in this study [39].

## 6. Limitations

The present study has some limitations that should be noted. First, this study used a retrospective design representing the experience of only two centers with a small size, relying solely on data accessed through the hospital information system. Additionally, due to its retrospective nature, it was able to establish associations but not causality. Second, the relatively small sample size may render this study underpowered in terms of detecting the value of the NPAR in predicting one-year MACCEs. Third, this study focused only on the NPAR levels at admission and ignored time-varying changes in the NPAR, which may provide more valuable information than the baseline NPAR for understanding the underlying mechanisms. Fourth, despite our efforts to comprehensively adjust for several important covariates, other unknown factors (e.g., problems in compliance with treatment) may have been present and introduced bias. Fifth, the findings obtained in the current study are exploratory and should be considered as hypothesis-generating. Therefore, further well-designed, prospective multi-center studies are needed to confirm and expand the results of our study.

## 7. Conclusions

Both the neutrophil percentage and albumin are easily available, cheap and useful in clinical practice. The NPAR, combined with these two factors, can be used as a promising predictor for risk stratification in NSTEMI patients in order to make a more precise decision regarding the therapeutic strategy and allocation of medical resources. However, further research on the predictive role of the NPAR and whether it can be combined with other symptom scores or traditional biomarkers to improve its predictive ability is required.

## Figures and Tables

**Figure 1 medicina-60-02101-f001:**
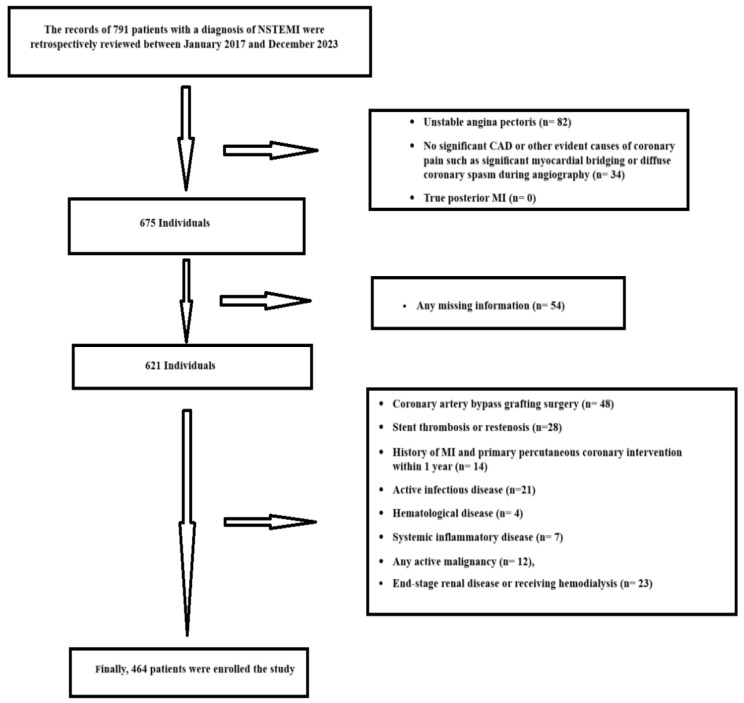
The study flow chart for the systematic selection method.

**Figure 2 medicina-60-02101-f002:**
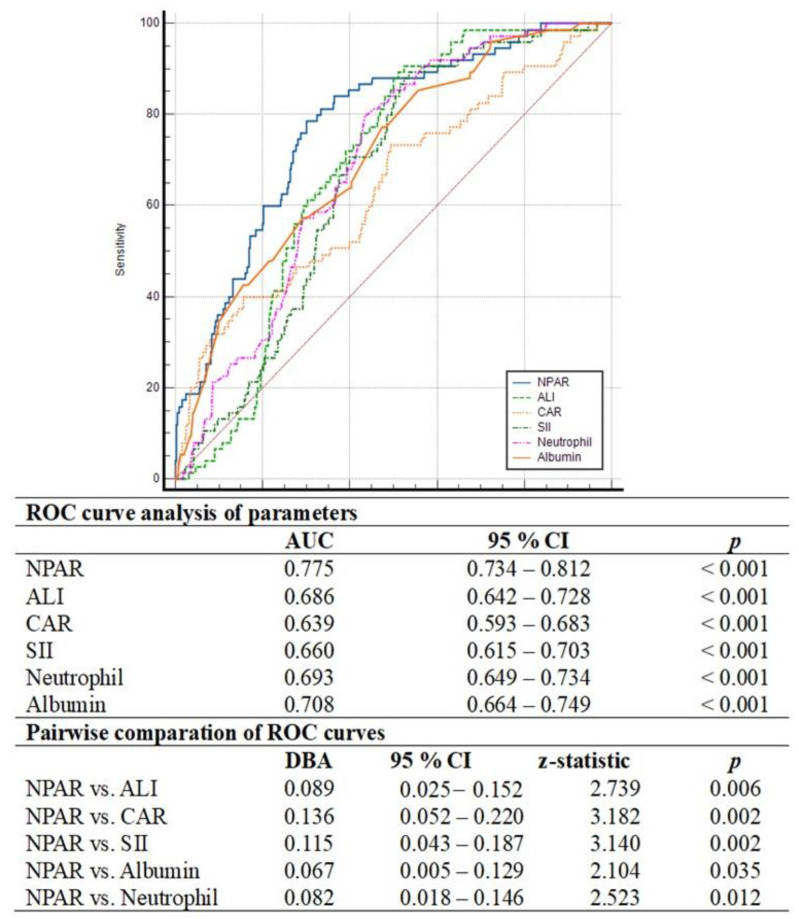
Comparison of predictive performances of NPAR, ALI, CAR, SII index, albumin and neutrophil determined by ROC curves in predicting long-term MACCEs. Abbreviations: ROC, receiver operating characteristic; AUC, area under the curve; CI, confidence interval; NPAR, neutrophil percentage-to-albumin ratio; ALI, advanced lung cancer inflammation index; CAR, C-reactive protein-to-albumin ratio; SII, systemic immune-inflammation; DBA, difference between areas.

**Figure 3 medicina-60-02101-f003:**
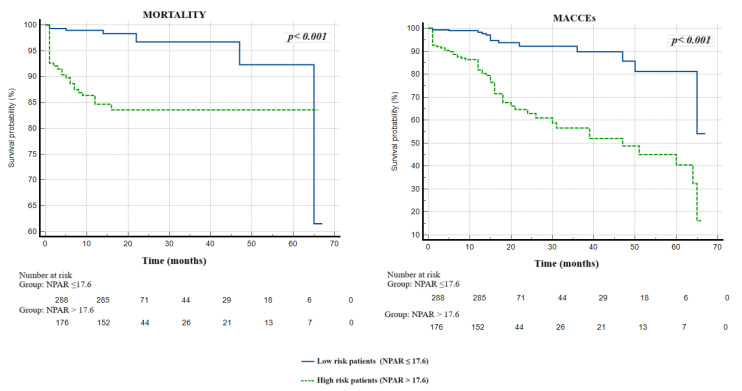
The Kaplan–Meier plots of the survival curves of patients with low (blue line) and high (green line) NPAR value categories. Abbreviations: NPAR, neutrophil percentage-to-albumin ratio; MACCEs, major adverse cardiovascular and cerebrovascular events.

**Table 1 medicina-60-02101-t001:** Demographical and clinical parameters of the study cohort grouped according to the presence of MACCEs.

Variables	All Population (n = 464)	Non-MACCEs (n = 389; 83.8%)	MACCEs (n = 75; 16.2%)	*p*
Male gender, n %	367 (79.1)	307 (78.9)	60 (80.0)	0.833
Age	59.2 ± 11.6	59.2 ± 11.7	57.8 ± 11.7	0.339
BMI (kg/m^2^)	27.4 ± 3.0	27.4 ± 3.0	27.2 ± 3.0	0.638
Hypertension, n (%)	232 (50.0)	196 (50.4)	36 (48.0)	0.705
Diabetes, n (%)	124 (26.7)	108 (27.8)	16 (21.3)	0.249
Hyperlipidemia, n (%)	168 (36.2)	130 (33.4)	38 (50.7)	0.004
Smoking, n (%)	207(44.6)	180 (46.3)	27 (36.0)	0.101
Family history, n (%)	193 (41.6)	163 (41.9)	30 (40.0)	0.760
CAD history, n (%)	136 (29.3)	114 (29.3)	22 (29.3)	0.996
PAD history, n (%)	25 (5.4)	23 (5.9)	2 (2.7)	0.254
Heart failure, n (%)	32 (6.9)	28 (7.2)	4 (5.3)	0.560
CRF, n (%)	33 (7.1)	25 (6.4)	8 (10.7)	0.191
Killip III-IV, n (%)	26 (5.6)	19 (4.9)	7 (9.3)	0.125
GRACE score	104.0 ± 24.1	102.7 ± 24.4	111.2 ± 21.2	0.005
LVEF (%)	48.4 ± 7.7	48.8 ± 7.6	46.4 ± 7.7	0.014
Medications, n (%)				
Acetylsalicyclic acid	159 (34.3)	135 (34.7)	24 (32.0)	0.651
ADP blockers	18 (3.9)	15 (3.9)	3 (4.0)	0.956
Beta-blockers	119 (25.6)	102 (26.2)	17 (22.7)	0.519
RAS blockers	174 (37.5)	153 (39.3)	21 (28.0)	0.063
CCBs, n (%)	103 (22.2)	87 (22.4)	16 (21.3)	0.844
Statin	56 (12.1)	47 (12.1)	9 (12.0)	0.984
Antianginals, n (%)	27 (5.8)	25 (6.4)	2 (2.7)	0.203
OADs, n (%)	105 (22.6)	92 (23.7)	13 (17.3)	0.231
Insulin, (%)	33 (7.1)	28 (7.2)	5 (6.7)	0.870
Syntax score I	12.0 [8.0–21.5]	12.0 [8.0–19.8]	19.0 [12.0–26.5]	<0.001
Syntax score II—PCI	25.0 [19.0–32.2]	23.6 [18.7–38.4]	31.3 [25.0–44.4]	<0.001
Nonfatal MI	18 (3.9)	0.0 (0.0)	18 (24)	<0.001
* Rehospitalization, CV-caused	38 (8.2)	0.0 (0.0)	38 (50.7)	<0.001
Stroke, n (%)	2 (0.4)	0.0 (0.0)	2 (2.7)	0.001
Mortality	35 (7.5)	0.0 (0.0)	35 (46.7)	<0.001
Require of revascularization, n (%)	32 (6.9)	0.0 (0.0)	32 (42.7)	<0.001

* Decompansated heart failure, unstable angina; hemodynamically significant ventricular arrhythmia. Abbreviations: BMI, body mass index; CAD, coronary artery disease; PAD, peripheral arterial disease; CRF, chronic renal failure; GRACE, Global Registry of Acute Coronary Events; LVEF, left ventricular ejection fraction; ADP, adenosine diphosphate; RAS, renin-angiotensin system; CCBs, calcium channel blockers; OADs, oral antidiabetic drugs; CV, cardiovascular.

**Table 2 medicina-60-02101-t002:** Laboratory parameters of the study cohort grouped according to the presence of MACCEs.

Variables	All Population (n = 464)	Non-MACCEs (n= 389; 83.8%)	MACCEs (n = 75; 16.2%)	*p*
FBG, (mg/dL)	122.0 [102.0–162.3]	120.0 [101.0–167.0]	124.0 [102.0–146.0]	0.342
eGFR	87.7 ± 22.1	88.2 ± 21.8	85.5 ± 23.9	0.345
Uric acid	5.36 ± 1.5	5.33 ± 1.5	5.48 ± 1.6	0.403
Albumin	3.73 ± 0.46	3.79 ± 0.45	3.46 ± 0.38	<0.001
CRP, IQR	5.30 [3.43–8.93]	5.10 [3.30–8.55]	6.50 [4.00–12.20]	<0.001
Troponin I, ng/L	210.0 [98.25–490.0]	199.0 [95.0–450.0]	280.0 [110.0–761.0]	0.040
CAR, IQR	1.41 [0.91–2.41]	1.33 [0.88–2.28]	1.76 [1.24–3.73]	<0.001
TC, mg/dL	207.4 ± 47.6	204.7 ± 47.9	221.4 ± 43.2	0.005
LDL-C, mg/dL	132.3 ± 39.9	129.4 ± 39.1	147.0 ± 41.1	<0.001
HDL-C, mg/dL	40.0 ± 11.0	40.1 ± 11.0	39.6 ± 10.7	0.757
Triglyceride, mg/dL	165.0 [120.0–233.8]	166.0 [119.5–232.5]	157.0 [120.0–241.0]	0.962
Hemoglobin	13.8 ± 1.7	13.7 ± 1.7	13.9 ± 1.9	0.403
WBC	8.98 ± 2.1	8.90 ± 2.1	9.30 ± 2.3	0.207
Neutrophil	5.31 ± 1.6	5.14 ± 1.6	6.14 ± 1.2	<0.001
Neutrophil percentage, %	59.7 ± 13.6	58.1 ± 13.5	68.3 ± 10.2	<0.001
Lymphocyte	1.79 [1.26–2.52]	1.80 [1.25–2.60]	1.70 [1.30–2.10]	0.131
Platelet	247.4 ± 69.7	245.0 ± 71.5	259.6 ± 57.9	0.096
SII index	694.7 [434.2–1108.1]	633.7 [407.7–1075.7]	899.2 [671.2–1232.6]	0.008
ALI, IQR	34.8 [22.9–52.5]	37.7 [24.3–57.1]	24.5 [20.8–33.8]	<0.001
NPAR	16.3 ± 4.2	15.6 ± 3.9	19.7 ± 3.8	<0.001

Abbreviations: FBG, fasting blood glucose; eGFR, estimated glomerular filtration rate; CRP, C-reactive protein; CAR, C-reactive protein-to-albumin ratio; TC, total cholesterol, LDL-C, low-density lipoprotein cholesterol, HDL-C, high-density lipoprotein cholesterol; WBC, white blood count; SII, systemic immune-inflammation; ALI, advanced lung cancer inflammation index; NPAR, neutrophil percentage-to-albumin ratio.

**Table 3 medicina-60-02101-t003:** Univariate cox regression analysis of all parameters and multivariate cox regression analysis (Model 1) of factors independently associated with MACCEs without inflammatory-based scores.

Variables	Univariate		Model 1 Multivariate	
	HR (95% CI)	*p*	HR (95% CI)	*p*
LVEF	0.972 (0.945–0.999)	0.039	0.983 (0.951–1.016)	0.313
GRACE	1.010 (1.002–1.019)	0.021	1.005 (0.995–1.015)	0.355
Syntax score I	1.061 (1.037–1.085)	<0.001	1.067 (1.041–1.094)	<0.001
LDL-C	1.011 (1.005–1.016)	<0.001	1.007 (1.002–1.013)	0.010
CRP	1.083 (1.037–1.131)	<0.001	1.025 (0.979–1.073)	0.286
Albumin	0.238 (0.142–0.402)	<0.001	0.206 (0.112–0.380)	<0.001
Neutrophil	1.404 (1.223–1.611)	<0.001	1.524 (1.298–1.788)	<0.001
Troponin I	1.001 (1.000–1.001)	0.039	1.000 (0.999–1.000)	0.291
ALI	0.965 (0.950–0.980)	<0.001	-	-
CAR	1.308 (1.154–1.484)	<0.001	-	-
SII index	1.000 (1.000–1.001)	0.006	-	-
NPAR	1.216 (1.153–1.284)	<0.001	-	-

Abbreviations: LVEF, left ventricular ejection fraction; GRACE, Global Registry of Acute Coronary Events; LDL-C, low-density lipoprotein cholesterol; CRP, C-reactive protein; ALI, advanced lung cancer inflammation index; CAR, C-reactive protein-to-albumin ratio; SII, systemic immune-inflammation; NPAR, neutrophil percentage-to-albumin ratio.

**Table 4 medicina-60-02101-t004:** Factors that were found to be independently associated with the MACCEs in multivariate cox regression analysis models including inflammation-based scores (Models 2 and 3).

Variables	Model 2		Model 3		Model 4	
	HR (95% CI)	*p*	HR (95% CI)	*p*	HR (95% CI)	*p*
LVEF	0.982 (0.953–1.012)	0.241	0.985 (0.955–1.017)	0.351	0.994 (0.963–1.026)	0.706
GRACE	1.005 (0.996–1.015)	0.288	1.006 (0.997–1.015)	0.204	1.004 (0.995–1.013)	0.428
SxS I	1.052 (1.027–1.078)	<0.001	1.063 (1.038–1.089)	< 0.001	1.061 (1.036–1.087)	<0.001
LDL-C	1.008 (1.003–1.014)	0.004	1.010 (1.004–1.016)	0.001	1.010 (1.004–1.016)	0.001
CRP	1.027 (0.982–1.075)	0.244	-	-	1.054 (1.007–1.103)	0.024
Troponin I	1.000 (0.999–1.000)	0.780	1.000 (1.000–1.001)	0.299	1.000 (1.000–1.001)	0.532
Albumin	-	-	-	-	-	-
Neutrophil	-	-	-	-	-	-
NPAR	1.184 (1.110–1.262)	<0.001	-	-	-	-
CAR	-	-	1.219 (1.068–1.393)	0.003	-	-
SII index	-	-	1.000 (1.000–1.001)	0.336	-	-
ALI					0.971 (0.956–0.987)	<0.001

Abbreviations: LVEF, left ventricular ejection fraction; GRACE, Global Registry of Acute Coronary Events; SxS I, Syntax score I; LDL-C, low-density lipoprotein cholesterol; CRP, C-reactive protein; NPAR, neutrophil percentage-to-albumin ratio; ALI, advanced lung cancer inflammation index; CAR, C-reactive protein-to-albumin ratio; SII, systemic immune-inflammation.

## Data Availability

The original contributions presented in the study are included in the article, further inquiries can be directed to the corresponding authors.

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
