# Peer review of "Prognostic Role of Neutrophil Percentage-to-Albumin Ratio in Patients with Non-ST-Elevation Myocardial Infarction"

_medicina, 2024, doi:10.3390/medicina60122101_

Round 1
Reviewer 1 Report
Comments and Suggestions for Authors
In the current study, the authors aimed to show an association between various pro-inflammatory or inflammatory markers or indexes and MACCE in patients with NSTEMI who underwent coronary angiography and PCI. The study was retrospective, and the sample size and MACCE rates were acceptable. The methodology was well-defined. Please remove the statement of the last sentence in the results section of the abstract (about SUA/SCr). On page 9, lines 295-296, there was a repetition of the neutrophil-albumin statement. How did the authors explain the last part of the MACCE Kaplan-Meier curve survival figure? There was no data about the compliance of the patients to the discharge medications, particularly for statins and antithrombotic drugs. Please include the peak CK-MB and Troponin values, their predictive value for MACCE, and their correlation with NPAR. Please also discuss the temporal variability of the NPAR rather than measuring it only at admission. I
Author Response
Dear Editor;
This letter contains our reply to reviewers’ suggestions about our manuscript entitled “Prognostic Role of Neutrophil Percentage-to-Albumin Ratio in Patients with Non-ST-Elevation Myocardial Infarction”. I and my colleague wish to thank you for your kindness that allowed us to reply your reviewer’s comments. This revision contains answers to the points underlined by reviewers, along with the revised manuscript. Changes in manuscript were highlighted with red.
Yours Sincerely,
Reviewer #1: We thank reviewer #1 for his/her constructive comments.
Q1: Please remove the statement of the last sentence in the results section of the abstract (about SUA/SCr).
A1: The written error at the last sentence of the abstract corrected according to your suggestion.
Q2: On page 9, lines 295-296, there was a repetition of the neutrophil-albumin statement.
A2: The correction on page 9, lines 295-296 has been made. Neutrophil to albumin ratio is the ratio of neutrophil to albumin numerically and as not the same as NPAR.
Q3: How did the authors explain the last part of the MACCE Kaplan-Meier curve survival figure?
A3: Kaplan-Meier curve reanalyzed and the sentence ‘Kaplan Meier curve indicate that high-risk patients with NPAR values ​​>17.6 have a significantly worse prognosis and adverse events during the follow-up period than the low-risk group’ added under the figure-3.
Q4: There was no data about the compliance of the patients to the discharge medications, particularly for statins and antithrombotic drugs.
A4: All the patients discharged with optimal medical treatment according to the recent guidelines. Compliance to treatment was asked at routine follow up visits and also by phone interviews to all participants. However, since there are possible problems in this regard, your opinion in this issue is justified therefore, the sentence at the limitations section ‘Fourth, despite our effort to comprehensively adjust for several important covariates, other unknown factors may be present and introduce bias’ changed as ‘Fourth, despite our efforts to comprehensively adjust for several important covariates, other unknown factors such as problems in compliance with treatment may be present and introduce bias’.
Q5: Please include the peak CK-MB and troponin values, their predictive value for MACCE, and their correlation with NPAR.
A5: Since CK-MB value is not evaluated both for diagnosis and follow-up we were not able to assess for this study. However, the recommended analysis regarding peak troponin levels have been made and the relevant corrections are indicated in red at the result section. Troponin values were evaluated in the univariate as well as multivariate regression analysis and showed a weak correlation with NPAR and no correlation with other two inflammation based parameters. Corrections made both in the text and in the regression analysis table are marked in red. In addition the correlation analysis method is also indicated as red in the statistical analysis section.
Q6: Please also discuss the temporal variability of the NPAR rather than measuring it only at admission.
A6: It was stated that in our study only NPAR values at admission was measured, however temporal variability was not evaluated which was an important limitation of our study.
Reviewer 2 Report
Comments and Suggestions for Authors
We would like to express our gratitude to the authors for submitting their manuscript titled "Prognostic Role of Neutrophil Percentage-to-Albumin Ratio in Patients with Non-ST Elevated Myocardial Infarction" to our journal. The topic is of significant interest, and the results are compellingly presented. However, we have outlined several suggestions to enhance the quality of the manuscript.
First, we recommend the creation of a graphical abstract that succinctly summarizes the main findings and messages of the article. This addition could improve clarity for readers and provide a quick reference to the study's key outcomes.
- Second, we suggest incorporating a discussion of the results in light of the article by Trimarchi et al. (PMID: 39458009 PMCID: PMC11508711), which examines the Advanced Lung Cancer Inflammation Index in the context of patients with ST-Elevation Myocardial Infarction (STEMI). This comparison may offer valuable insights and contextualize the authors' findings within the broader field of myocardial infarction research.
Furthermore, it would be beneficial to include the number of patients at risk within the Kaplan-Meier curves to enhance interpretability. Lastly, we recommend assessing the impact of the Advanced Lung Cancer Inflammation Index ((albumin x BMI) / Neutrophil-Lymphocyte Ratio) in the study population and comparing it with the proposed Neutrophil Percentage-to-Albumin Ratio (NPAR). These revisions are intended to strengthen the manuscript's contributions to the literature.
Author Response
Dear Editor;
This letter contains our reply to reviewers’ suggestions about our manuscript entitled “Prognostic Role of Neutrophil Percentage-to-Albumin Ratio in Patients with Non-ST-Elevation Myocardial Infarction”. I and my colleague wish to thank you for your kindness that allowed us to reply your reviewer’s comments. This revision contains answers to the points underlined by reviewers, along with the revised manuscript. Changes in manuscript were highlighted with red.
Yours Sincerely,
Reviewer #2: We thank reviewer #2 for his/her constructive comments.
Q1: First, we recommend the creation of a graphical abstract that succinctly summarizes the main findings and messages of the article. This addition could improve clarity for readers and provide a quick reference to the study's key outcomes.
A1: Graphical abstract has been created and uploaded to system according to reviewer’s suggestion.
Q2: Second, we suggest incorporating a discussion of the results in light of the article by Trimarchi et al. (PMID: 39458009 PMCID: PMC11508711), which examines the Advanced Lung Cancer Inflammation Index in the context of patients with ST-Elevation Myocardial Infarction (STEMI). This comparison may offer valuable insights and contextualize the authors' findings within the broader field of myocardial infarction research. Furthermore, it would be beneficial to include the number of patients at risk within the Kaplan-Meier curves to enhance interpretability. Lastly, we recommend assessing the impact of the Advanced Lung Cancer Inflammation Index ((albumin x BMI) / Neutrophil-Lymphocyte Ratio) in the study population and comparing it with the proposed Neutrophil Percentage-to-Albumin Ratio (NPAR). These revisions are intended to strengthen the manuscript's contributions to the literature.
A2: In line with your suggestions ALI index evaluated and added to our manuscript as well as the article by Trimarchive et al. was used.
Reviewer 3 Report
Comments and Suggestions for Authors
The authors evaluated the prognostic significance of the neutrophil percentage-to-albumin ratio (NPAR) in NSTEMI patients undergoing PCI in case of major adverse cardiovascular and cerebrovascular events (MACCE) at one year from the index event.
The study has many limitations, of which the authors are aware, first of all the retrospective design and the relatively small sample size that makes it underpowered for the purpose of the study; nevertheless, it offers interesting insights and seems to highlight a better predictive role of NPAR compared to other inflammatory markers whose predictive role has already been demonstrated in this clinical context.
Some suggestions for the authors:
a review of the English text is needed;
line 32: define SUA/SCr
line 82: insert (PAD), i.e. the acronym used subsequently
lines 90-92: indicate the target level and cite the source
lines 99-100: define the "kidney damage" (assessed with imaging methods? other?)
lines 120-125 and 132-139: since these are data from retrospectively reviewed medical records, omit non-applicable details and assessment methods (method of blood sampling, vein used, assessment of angiograms, calculation of LVEF, etc.)
table 1: eliminate the statistical comparison between the Non-MACCE group and the MACCE group for individual MACCEs: this is precisely the difference that distinguishes the two groups.
line 364: after "further research is required on" also add "the predictive role of NPAR and", to deemphasize findings that cannot be interpreted as definitive due to limitations of the study.
Comments on the Quality of English LanguageA review of the English text is needed.
Author Response
Dear Editor;
This letter contains our reply to reviewers’ suggestions about our manuscript entitled “Prognostic Role of Neutrophil Percentage-to-Albumin Ratio in Patients with Non-ST-Elevation Myocardial Infarction”. I and my colleague wish to thank you for your kindness that allowed us to reply your reviewer’s comments. This revision contains answers to the points underlined by reviewers, along with the revised manuscript. Changes in manuscript were highlighted with red.
Yours Sincerely,
Reviewer #3: We thank reviewer #3 for his/her constructive comments.
Q1: A review of the English text is needed
A1: Professional support has been received for English editing and the certificate has been uploaded.
Q2: Line 32: define SUA/SC
A2: "Individuals with a SUA/SCr value greater than 4.58 had a statistically increased risk for developing MACCEs (p< 0.001)" in Line 32 belongs to another article we are preparing and was included here by mistake. It has been corrected and written in red.
Q3: Line 82: insert (PAD), i.e. the acronym used subsequently
A3: Suggested corrections are written and highlighted in red.
Q4: lines 90-92: indicate the target level and cite the source
A4: Corrections regarding the target level and citations have been made and written in red. The CRF definition has been corrected more clearly and the source is shown. Corrections are shown in red.
Q5: lines 120-125 and 132-139: since these are data from retrospectively reviewed medical records, omit non-applicable details and assessment methods (method of blood sampling, vein used, assessment of angiograms, calculation of LVEF, etc.)
A5: In line with your suggestions, arrangements have been made with the methodology of the laboratory parameters. However, since the angiographic images of the patients are available in our system and the analyses are performed by viewing the images again, no changes have been made in the part related to SxSI.
Q6: line 364: after "further research is required on" also add "the predictive role of NPAR and", to deemphasize findings that cannot be interpreted as definitive due to limitations of the study.
A6: In line with your valuable suggestions, line 364 has been corrected and marked in red.
Reviewer 4 Report
Comments and Suggestions for Authors
To the Authors,
The present study investigated the predictive role of various inflammatory and biochemical blood markers, collected at the time of admission, in determining the 1-year risk of major adverse cardiac and cerebrovascular events (MACCEs) in patients with non-ST segment elevation myocardial infarction (NSTEMI). Among the analysed biomarkers, the neutrophil percentage-to-albumin ratio (NPAR) and serum albumin emerged as stronger predictors of NSTEMI-associated MACCEs compared to other markers, such as the CRP-to-albumin ratio (CAR) and the systemic immune-inflammation (SII) index. Notably, NPAR demonstrated superior predictive accuracy for the development of MACCEs in these patients. Moreover, anatomical assessment through SYNTAX scoring further revealed that NSTEMI patients who experienced MACCEs within one year were more likely to have complex coronary artery disease than those without such complications.
These findings underscore the importance of simple, cost-effective, and accessible inflammatory blood biomarkers-particularly NPAR-in predicting cardiac and cerebrovascular complications. Such biomarkers could serve as valuable adjuncts in improving long-term outcomes and survival rates in patients with NSTEMI.
The authors are commended for presenting these novel findings on the role of inflammatory blood biomarkers in predicting MACCEs in NSTEMI patients. However, the following comments need to be addressed before the manuscript can be accepted for publication.
Best regards,
Major Comments:
· English language issues: Please see below.
· Methods and materials:
- The inclusion and exclusion criteria are not clearly identified or poorly described. The lines 103-111 provide a poor and lengthy description of the conditions that were excluded from the study. To clarify this, the authors can provide a clear Flow Chart or diagram where they can provide a nice flow of the entire number of the participants and then to be followed by inclusion and exclusion conditions.
- Definition of the clinical conditions: the authors need to support their definition of diabetes mellitus, hypertension, hyperlipidaemia and PDA in the participants with common clinical guidelines, namely AHA, WHO, NICE, etc. In addition, it is not clear why ‘using anti-diabetic medications’ or ‘anti-hypertensive medications’ were among the criteria used to define diabetes mellitus and hypertension, respectively.
- NSTEMI diagnosis: will there be any possible association between the levels of cardiac troponin (cTnT) and NPAR, for example, in predicating the MACCEs in the patients? Also, was there any incidence of posterior myocardial infarction in the participants and were these excluded from the study?
· Results:
- Line 176: the results seem to be confusion. Are the following conditions are too considered part of MACCEs? If so, the total number of the patients who developed MACCEs will be 90 instead of 75: nonfatal MI (n=18) + rehospitalization (n=38) + recurrent PCI (n=32) + stroke (n=2). Can the authors please elaborate on this?
- Stroke patients: it appears only two of the participants had developed stroke during the follow up period. The question here is: were there any patients with TIAs? As such, the results regarding the MACCEs and biomarkers in the patients need not to be exaggerated due to the small numbers of patients with cerebrovascular events.
- It appears from the Table 1 results, BMI, that the participants were overweight. In that aspect, did the participants have any liver abnormalities, such as through interpreting the liver function tests, that would need to be excluded from this study?
· Discussion: lines 336-344: it is really not clear why this argument has been placed at this very end section. The explanation is for justifying the overall aim of the study, and therefore, it is either repeated, implying to be removed; Or it is advised to be placed somewhere at the end of the introduction section rather than at the very end part of the discussion section.
Minor Comments:
· Line 2: ‘Elevated’ to be changed to ‘Elevation’.
· Line 32: it is not clear what is the exact meaning of ‘Individuals with a SUA/Scr’. Please clarify this.
· Line 33: Please add ‘A’ before ‘ Newly’.
· Line 35: please add ‘in’ before ‘NSTEMI patients.’.
· Line 60: the phrase needs to be properly referenced.
· Line 82: please provide the abbreviation of ‘PDA’ after ‘peripheral arterial disease’.
· Line 83: add ‘the’ before ‘…patients’.
· Lines 103-111: please see the comment in Major Comments section, too. Since the number comes at the beginning of the sentence, it needs to be spelled out (Seven hundreds…).
· Lines 128 and 129: the sentence starts with ‘. CAR and SII index… until also calculated.’ is extra which better to be removed.
· Lines 133 and 134: the sentence starts with ‘. Coronary angiograms were…until clinical data.’ is awkwardly repeated and needs to be removed.
· Line 146: it is indicated that the ‘informed consent’ was not obtained, but then in line 373 it is mentioned it was. This needs to be very properly and effectively addressed as it is related to the patients’ clinical data.
· Line 182: with regard to ‘cardiovascular risk factors..’ this is a very general description which needs to be either removed or properly addressed, such as by specifically elaborating on these risk factors, DM, HTN, etc.
· Line 194: the larger median result (1.76) needs to be versus median 1.3, and not the other way around.
· Line 225: the abbreviation ‘CV’ is not included in the Table, and thus, needs to be removed.
· Table 2: Glucose: the authors need to indicate whether this was a random or fasting blood glucose.
· Figure 2: left hand Figure: the tick marks on the Y axis are very crowdy, and hence, both figures need to have an equal amount/distance of tick marks.
· Line 282: please add ’ as’ before ‘… an indicator of nutritional…’.
· Line 309: please add ‘a’ before ‘… better predictive…’.
· Line 319: please replace ‘we think…’ with ‘we suspect…’.
· Line 333: please put the citation number 33 after ‘(lipid paradox)’.
· Line 353: please add ‘ the’ before … baseline’.
Comments on the Quality of English Language
· English language issues: To begin with, it is acknowledged that English may not be the authors' first language, and their efforts to present this work in English are highly commendable. However, the manuscript contains repeated grammatical and language errors that require significant revision. Therefore, addressing these issues is essential as part of the Major Comments. The manuscript should undergo thorough proofreading and editing to improve its clarity and readability prior to resubmission. Among these issues:
1) Punctuation issues:
- Line 3: the second full stop that follows ‘M.D’ needs to be removed.
- Line 16: the initial (open) parenthesis before MACCEs needs to be added.
- Line 33: a comma needs to be added after ‘NPAR’.
- Line 63: a comma needs to be added before ‘such as’.
- Line 64: a comma needs to be added before ‘stroke’.
- Line 81: a comma needs to be added after ‘vascular diseases’.
- Line 326: the semicolon after ‘…in our study’ needs to be removed.
2) Sentence structure and other grammatical errors: in a few places, the sentences are incorrectly established, which results in an overall vague meaning of the sentence. Moreover, there is an incorrect usage of the ‘to be’ verbs throughout the manuscript. For example:
- Line 21: ‘were’ needs to be changed to ‘was’.
- Line 22: ‘MACCE was’ needs to be replaced with a plural form: MACCEs were.
- Line 62: ‘was’ after ‘interval’ needs to be replaced with ‘were’.
- Lines 33-35: the conclusion needs to be written in a more concise way.
- Lines 64-66: there is an incorrect usage of the past tense to establish the sentence that starts with ‘..However,…’ and ends with ‘… of NSTEMI patients’. Instead, the sentence can be re-written using the present perfect tense (..have been no studies..).
- Lines 66-69: the sentence starts with ‘ Currently…’: again, this is a poorly and lengthy written section that needs to be re-written in a shorter and clearer way.
- Line 75: the date (between January 2017…) needs to be placed at the end of the sentence after ‘reviewed’.
- Line 81: please add ‘… ,which were’ after ‘vascular diseases’.
- Lines 150 and 151: ‘(if a normal distribution)’ again, this needs to be rephrased in a better way.
- Line 277: ‘inflammation’ needs to be changed to a plural form and then followed by ‘were’.
- Lines 292-294: the sentence that stars with ‘Indeed..’ and ends with ‘inflammation.’ is very awkwardly established which needs to be replaced.
- Lines 322-324: again, the sentence (In a study…. predicting the survival.) does not make sense and needs to be replaced.
- Line 347: ‘center’ needs to be plural.
- Lines 348-349: please rephrase the sentence (,which was able….bias have been introduced) to a clear and more meaningful phrase.
Author Response
Reviewer #4: We thank reviewer #4 for his/her constructive comments.
Q1: The inclusion and exclusion criteria are not clearly identified or poorly described. The lines 103-111 provide a poor and lengthy description of the conditions that were excluded from the study. To clarify this, the authors can provide a clear Flow Chart or diagram where they can provide a nice flow of the entire number of the participants and then to be followed by inclusion and exclusion conditions.
A1: A study flow chart has been drawn to describe the exclusion criteria more precisely according to your suggestion.
Q2: Definition of the clinical conditions: the authors need to support their definition of diabetes mellitus, hypertension, hyperlipidaemia and PDA in the participants with common clinical guidelines, namely AHA, WHO, NICE, etc. In addition, it is not clear why ‘using anti-diabetic medications’ or ‘anti-hypertensive medications’ were among the criteria used to define diabetes mellitus and hypertension, respectively.
A2: Definitions have been made essentially iteratively as recommended in treatment guidelines.
Q3: NSTEMI diagnosis: will there be any possible association between the levels of cardiac troponin (cTnT) and NPAR, for example, in predicating the MACCEs in the patients? Also, was there any incidence of posterior myocardial infarction in the participants and were these excluded from the study?
A3: The association with troponin levels and NPAR was evaluated and added. True posterior MI was not detected in any of the patients, and those with a posterior MI diagnosis were added to the exclusion criteria in line with your suggestions.
Q4: Line 176: the results seem to be confusion. Are the following conditions are too considered part of MACCEs? If so, the total number of the patients who developed MACCEs will be 90 instead of 75: nonfatal MI (n=18) + re-hospitalization (n=38) + recurrent PCI (n=32) + stroke (n=2). Can the authors please elaborate on this?
A4: We determined that the main reason for the discrepancy between the numbers was that nonfatal MI patients were also coded as re-hospitalization and we corrected this situation that we overlooked. In addition, in line with your attention and valuable suggestions, our data and text were reviewed one by one and we realized that the sentence of cardiovascular re-hospitalization, which constitutes our endpoints, was not written and we changed our endpoint definition to "The primary endpoint of the present study was MACCEs, defined as cardiovascular death (comprising myocardial infarction, significant cardiac arrhythmia, heart failure and any stroke), nonfatal myocardial infarction, nonfatal cerebrovascular accident (ischemic stroke or TIA) or cardiovascular re-hospitalization due to myocardial infarction, significant cardiac arrhythmia, heart failure and any stroke within the 12 months follow-up period'. The result part has been rewritten to make it more understandable and simpler, based on the definition. The section on re-hospitalization due to CV has also been corrected in the table.
Q5: Stroke patients: it appears only two of the participants had developed stroke during the follow up period. The question here is: were there any patients with TIAs? As such, the results regarding the MACCEs and biomarkers in the patients need not to be exaggerated due to the small numbers of patients with cerebrovascular events.
A5: Both patients developed a stroke however there was no detection of TIA.
Q6: It appears from the Table 1 results, BMI, that the participants were overweight. In that aspect, did the participants have any liver abnormalities, such as through interpreting the liver function tests that would need to be excluded from this study?
A6: None of the patients included in the study had known liver disease. Unfortunately, the rate of overweight or obesity in the Turkish population is higher than in European countries due to unhealthy eating habits, lack of exercise and sedentary lifestyle. The article I present to you below supports these findings.
Reference: Seri A, Arslan N. Obesity in adults in Turkey: age and regional effects. Eur J Public Health. 2009 Jan;19 (1):91-4. doi: 10.1093/eurpub/ckn107.
Q7: lines 336-344: it is really not clear why this argument has been placed at this very end section. The explanation is for justifying the overall aim of the study, and therefore, it is either repeated, implying to be removed; Or it is advised to be placed somewhere at the end of the introduction section rather than at the very end part of the discussion section.
A7: According to your suggestions, corrections were made and the relevant part was removed from the discussion section and added to the introduction part.
Q8: Line 2: ‘Elevated’ to be changed to ‘Elevation’.
A8: The correction has been made in line with your suggestion.
Q9: Line 32: it is not clear what is the exact meaning of ‘Individuals with a SUA/Scr’. Please clarify this.
A9: Line 32 is a misspelling and has been removed from the text.
Q10: Line 33: Please add ‘A’ before ‘ Newly’
A10: The correction has been made in line with your suggestion.
Q11: Line 35: please add ‘in’ before ‘NSTEMI patients.’
A11: The correction has been made in line with your suggestion.
Q12: Line 60: The phrase needs to be properly referenced.
A12: A reference has been added according to your suggestion.
Q13: Line 82: please provide the abbreviation of ‘PDA’ after ‘peripheral arterial disease’.
A13: The correction has been made in line with your suggestion.
Q14: Line 83: add ‘the’ before ‘…patients’.
A14: The correction has been made in line with your suggestion.
Q15: Lines 103-111: please see the comment in Major Comments section, too. Since the number comes at the beginning of the sentence, it needs to be spelled out (Seven hundreds…).
A15: The correction has been made in line with your suggestion.
Q16: Lines 128 and 129: the sentence starts with ‘. CAR and SII index… until also calculated.’ is extra which better to be removed.
A16: The sentence ‘’CAR and SII were also calculated’’ was removed from the text. In line with the recommendations, the Advanced Lung Cancer Inflammation Index (ALI) calculation was included.
Q17: Lines 133 and 134: the sentence starts with ‘. Coronary angiograms were…until clinical data.’ is awkwardly repeated and needs to be removed.
A17: The correction has been made in line with your suggestion.
Q18: Line 146: it is indicated that the ‘informed consent’ was not obtained, but then in line 373 it is mentioned it was. This needs to be very properly and effectively addressed as it is related to the patients’ clinical data.
A18: The study was a retrospective analysis and written consent was not obtained. The phrase in Line 373 was marked as incorrect in the system during the uploading process of the article.
Q19: Line 182: with regard to ‘cardiovascular risk factors..’ this is a very general description which needs to be either removed or properly addressed, such as by specifically elaborating on these risk factors, DM, HTN, etc.
A19: Your suggestions have been corrected and the expression cardiovascular factors has been clarified.
Q20: Line 194: the larger median result (1.76) needs to be versus median 1.3, and not the other way around.
A20: The correction has been made in line with your suggestion.
Q21: Line 225: the abbreviation ‘CV’ is not included in the Table, and thus, needs to be removed.
A21: 'Re-hospitalization CV-caused' is included in the table. The phrase is highlighted in red.
Q22: Table 2: Glucose: the authors need to indicate whether this was a random or fasting blood glucose.
A22: Table 2. Glucose values ​​were taken from a fasting blood sample and the relevant part was changed to fasting blood glucose (FBG).
Q23: Figure 2: left hand Figure: the tick marks on the Y axis are very crowdy, and hence, both figures need to have an equal amount/distance of tick marks.
A23: Figure 2 has been revised in line with your suggestion.
Q24: Line 282: please add ’ as’ before ‘… an indicator of nutritional…’.
A24: The correction has been made in line with your suggestion.
Q25: Line 309: please add ‘a’ before ‘… better predictive…’.
A25: The correction has been made in line with your suggestion.
Q26: Line 319: please replace ‘we think…’ with ‘we suspect…’.
A26: The correction has been made in line with your suggestion.
Q27: Line 333: please put the citation number 33 after ‘(lipid paradox)’.
A27: Recommended edit has been made
Q28: Line 353: please add ‘ the’ before … baseline’.
A28: The correction has been made in line with your suggestion.
Q29: The manuscript should undergo thorough proofreading and editing to improve its clarity and readability prior to resubmission.
A29: Professional support has been received for English editing and the certificate has been uploaded.
Q30: Line 3: the second full stop that follows ‘M.D’ needs to be removed.
A30: The correction has been made in line with your suggestion.
Q31: Line 16: the initial (open) parenthesis before MACCEs needs to be added.
A31: The correction has been made in line with your suggestion.
Q32: Line 33: a comma needs to be added after ‘NPAR’.
A32: The correction has been made in line with your suggestion.
Q33: Line 63: a comma needs to be added before ‘such as’.
A33: The correction has been made in line with your suggestion.
Q34: Line 64: a comma needs to be added before ‘stroke’.
A34: The correction has been made in line with your suggestion.
Q35: Line 81: a comma needs to be added after ‘vascular diseases’.
A35: The correction has been made in line with your suggestion.
Q36: Line 326: the semicolon after ‘…in our study’ needs to be removed.
A36: The correction has been made in line with your suggestion.
Q37: Line 21: ‘were’ needs to be changed to ‘was’.
A37: The correction has been made in line with your suggestion.
Q38: Line 22: ‘MACCE was’ needs to be replaced with a plural form: MACCEs were.
A38: The correction has been made in line with your suggestion.
Q39: Lines 33-35: the conclusion needs to be written in a more concise way.
A39: Conclusion section has been revised in line with your recommendations.
Q40: Lines 64-66: there is an incorrect usage of the past tense to establish the sentence that starts with ‘..However,…’ and ends with ‘… of NSTEMI patients’. Instead, the sentence can be re-written using the present perfect tense (..have been no studies..)
A40: These sentences have been completely changed in line with your suggestions above.
Q41: Lines 66-69: the sentence starts with ‘ Currently…’: again, this is a poorly and lengthy written section that needs to be re-written in a shorter and clearer way.
A41: These sentences have been completely changed in line with your suggestions above.
Q42: Line 75: the date (between January 2017…) needs to be placed at the end of the sentence after ‘reviewed’.
A42: The correction has been made in line with your suggestion.
Q43: Line 81: please add ‘… ,which were’ after ‘vascular diseases’.
A43: The correction has been made in line with your suggestion.
Q44: Lines 150 and 151: ‘(if a normal distribution)’ again, this needs to be rephrased in a better way.
A44: These sections has been revised and rewritten in red.
Q45: Line 277: ‘inflammation’ needs to be changed to a plural form and then followed by ‘were’.
A45: The correction has been made in line with your suggestion.
Q46: Lines 292-294: the sentence that stars with ‘Indeed..’ and ends with ‘inflammation.’ is very awkwardly established which needs to be replaced.
A46: These sections has been revised and rewritten in red.
Q47: Lines 322-324: again, the sentence (In a study…. predicting the survival.) does not make sense and needs to be replaced.
A47: These sections has been revised and rewritten in red.
Q48: Line 347: ‘center’ needs to be plural.
A48: The correction has been made in line with your suggestion.
Q49: Lines 348-349: please rephrase the sentence (,which was able….bias have been introduced) to a clear and more meaningful phrase.
A49: These sections has been revised and rewritten in red.
Round 2
Reviewer 4 Report
Comments and Suggestions for Authors
To the Authors,
The authors are thanked for their follow up on the comments provided on the manuscript. The are specifically thanked for grammatically improving the writing language of the manuscript, which now appears polished and clear to the readers.
The authors should also be commended for incorporating a diagram that has organized the design of the manuscript, allowing the reader to smoothly follow up on the manuscript's contents. Lastly, including the troponin values and the advanced lung cancer inflammation index are important add-on factors to further specify the role of NPAR and MACCEs and rule out further confounding factors in the NSTEMI patients-thank you.
Sincerely,